# Interpretable Machine Learning for Explaining and Predicting Collapse Hazards in the Changbai Mountain Region

**DOI:** 10.3390/s25051512

**Published:** 2025-02-28

**Authors:** Xiangyang He, Qiuling Lang, Jiquan Zhang, Yichen Zhang, Qingze Jin, Jinyuan Xu

**Affiliations:** 1School of Jilin Emergency Management, Changchun Institute of Technology, Changchun 130012, China; hexiangyang@stu.ccit.edu.cn (X.H.); zhangyc@ccit.edu.cn (Y.Z.); holahola123@foxmail.com (Q.J.); xujinyuan@stu.ccit.edu.cn (J.X.); 2School of Environment, Northeast Normal University, Changchun 130117, China; zhangjq022@nenu.edu.cn

**Keywords:** collapse hazard assessment, machine learning, Changbai Mountain, SHAP

## Abstract

This study analyzes collapse hazards for complex interactions between geology, meteorology, and human activities in the Changbai Mountain region, focusing on how to cope with these features through machine learning. Using a dataset of 651 collapse events, this study evaluates four machine learning methods, Support Vector Machine (SVM), Random Forest (RF), Extreme Gradient Boosting (XGBoost), and Light Gradient Boosting Machine (LightGBM), to deal with complex nonlinear data structures. To overcome the limitations of a single-feature selection method, a variance inflation factor is introduced to optimize the selection of collapse risk factors. The transparency and interpretability of the modeling results are enhanced by combining the Shapley Additive Explanations (SHAP) with interpretable artificial intelligence. Model performance is evaluated on a test set by several statistical metrics, which shows that the optimized random forest model performs best and outperforms SVM, XGBoost, and LightGBM. The SHAP analysis results indicate that distance from the road is a key factor for collapse hazard. This study emphasizes the need for collapse management strategies that provide interpretable solutions for collapse hazard assessment.

## 1. Introduction

In recent years, the assessment of collapse hazards has gained increasing prominence within the field of geological disaster prevention [1,2]. Although progress has been made in understanding the mechanisms behind collapse formation, a further in-depth exploration of the complex causative processes remains essential. Machine learning technologies have demonstrated considerable potential in analyzing various nonlinear factors, such as geology, topography, and rainfall, and have been extensively applied in collapse hazard assessment [3]. However, existing machine learning models generally lack interpretability, which limits the deeper understanding of the relationships between collapse-triggering factors over large areas, thereby undermining the transparency and credibility of model predictions.

In the Changbai Mountain region, current collapse hazard assessment methods primarily focus on quantitatively describing the probabilities associated with various evaluation factors [4]. These methods often overlook the interaction mechanisms between factors, necessitating a further investigation into the spatial distribution patterns of collapse occurrence probabilities and the interactions among evaluation factors. Currently, collapse hazard assessment methods can be broadly classified into qualitative and quantitative approaches [5,6]. Qualitative methods rely on expert judgment to assign weights to evaluation factors [7], but they are prone to subjective bias and struggle to quantitatively analyze the interactions between factors. Quantitative methods include physical and statistical models; the former are limited by difficulties in obtaining comprehensive physical parameters [8], while traditional statistical methods, such as frequency ratios [9], evidence weighting [10], and analytic hierarchy processes [11], are constrained in their ability to handle complex nonlinear relationships, primarily capturing statistical regularities.

In contrast, machine learning models, such as decision trees [12], Support Vector Machines [13], and logistic regression [14], offer a certain degree of interpretability through their internal structures. However, these models still face challenges in effectively mining nonlinear relationships and identifying local features when dealing with complex geological issues. Advanced machine learning methods, such as neural networks [15], Random Forests [16], and Extreme Gradient Boosting [17], aim to enhance the accuracy of collapse hazard assessments by quantifying feature importance, but their interpretability remains insufficiently comprehensive, particularly in accurately reflecting the global behavior of models under varying geological conditions.

In recent years, researchers have placed increasing emphasis on enhancing the transparency of machine learning “black-box” models through interpretability methods [18]. Local Interpretable Model-agnostic Explanations [19] (LIME) and Shapley values [20] are among the most prominent interpretability tools. Although LIME is effective in some environments, it struggles with handling nonlinear models and high-dimensional data, and its explanations may be biased due to its excessive sensitivity to local input data. In contrast, the Shapley value method, originating from cooperative game theory, assigns quantifiable impacts to each feature, facilitating a more in-depth analysis of the model’s prediction process from local, global, and spatial perspectives [21].

The primary objective of this study is to generate accurate and interpretable collapse hazard maps for the Changbai Mountain region. To achieve this, we integrate advanced machine learning models with interpretability tools, such as Shapley values, to explore both local and global feature interactions. Specifically, we seek to answer the following questions: (1) How can we improve the interpretability of machine learning models for collapse hazard assessment? (2) What are the key factors influencing collapse hazard assessments in this region? (3) How can we quantify and optimize the interactions between geological, topographical, and environmental factors to enhance the accuracy of collapse hazard prediction? To achieve these objectives, we employ particle swarm optimization to fine-tune the hyperparameters of four widely used machine learning models: Random Forest (RF), Support Vector Machine (SVM), Extreme Gradient Boosting (XGBoost), and Light Gradient Boosting (LightGBM). This optimization will improve the predictive capabilities of the models and contribute to the development of more accurate and interpretable collapse hazard maps for the Changbai Mountain region.

## 2. Study Area

### Overview of the Study Area

The study area is located in the southeastern part of Jilin Province, with geographical coordinates ranging from 41°09′ to 43°24′ N latitude and 127°25′ to 128°56′ E longitude. This region encompasses Changbai, Antu, and Fusong counties, covering an approximate area of 16,080.8 km^2^. Situated in the heart of the Changbai Mountains, the area is characterized by mid- to high-mountain topography and serves as a major source for the Songhua and Yalu Rivers. Figure 1 illustrates the geographic location of the study area and the spatial distribution of historical collapse hazards. The region’s complex and varied geography, pronounced topographic relief, and frequent seismic activity contribute to its susceptibility to natural hazards.

Changbai County lies at the southern base of the Changbai Mountains in southeastern Jilin Province, on the right bank of the upper Yalu River, adjacent to Linjiang City, and bordering Fusong County to the north. The southeastern boundary of Fusong County is defined by the Yalu River, which also forms a natural border with North Korea. Similarly, Antu County, also located in southeastern Jilin Province, is separated from North Korea by the Yalu River. The region lies within the north temperate continental humid monsoon climate zone, characterized by distinct climatic features: long, cold winters and short, rainy summers. The pronounced seasonal climatic conditions significantly influence the occurrence and progression of collapse hazards in the region. Notably, heavy rainfall events associated with the monsoon climate significantly affect the spatial distribution and temporal evolution of collapse risks. The region’s complex topography, shaped by the volcanic activity of the Changbai Mountains, is further complicated by frequent geological phenomena, including seismic events and volcanic eruptions. These geological and geomorphological characteristics create a unique environment that is particularly prone to collapse hazards. The interplay between volcanic processes, seismic activity, and diverse topographical features makes the region an ideal case for studying natural hazard assessments [22]. Given these factors, conducting a comprehensive evaluation of collapse hazards is critical for understanding the risks and implementing effective disaster mitigation strategies. Figure 2 shows field photographs from the site investigation.

## 3. Data and Research Methods

### 3.1. Collapse Hazard Assessment Factors

Combining the causal mechanisms of collapse disasters with the specific characteristics of the Changbaishan Mountain area and referencing a large body of the literature, it is evident that the formation of collapse disasters is primarily influenced by various factors such as topography, geology, hydrology, meteorology, vegetation cover, and human engineering activities. In this study, twelve factors, including elevation, slope, slope direction, curvature, topographic humidity index, surface roughness, lithology, fault distance, road distance, river distance, the NDVI, and average annual rainfall, were selected for hazard evaluation. Table 1 presents the primary data sources.

#### 3.1.1. DEM and Derived Terrain Factors

Digital Elevation Models (DEMs) are essential in collapse hazard assessment. DEMs provide critical data on the topographical undulations of an area, which can subsequently be used to derive various topographic features such as slope, aspect, plan curvature, and profile curvature. This study employed ASTGTM2 DEM data with a 30 m resolution [23]. Following the standardization of the coordinate system, mosaicking, clipping, and the correction of anomalies in the original DEM using GIS software (ArcGIS 10.8), the following key topographic factors were extracted:

Elevation: Affects temperature and precipitation, which in turn influence soil moisture and, to some extent, the stability of geotechnical structures.

Slope: Directly impacts the likelihood of collapse occurrence, with steeper slopes indicating a higher potential for collapse speed and intensity [24].

Aspect: Controls the amount of solar radiation received by the slope surface, influencing soil moisture and, consequently, altering the likelihood of collapses [25].

Curvature: Indicates the degree of terrain convexity or concavity, influencing the accumulation of moisture and the distribution of vegetation; generally, convex slopes are more prone to instability [26].

Topographic Wetness Index (TWI): Calculated using parameters such as the catchment area and slope, reflecting the terrain’s capacity for moisture convergence and retention. Higher TWI values indicate a greater accumulation of moisture and higher soil moisture content [27], thereby increasing the potential risk of collapses.(1)$$TWI=\ln\left(\frac{A_{s}}{\tan\beta }\right)$$
where $A_{s}$ is the unit catchment area and *β* is the slope.i

#### 3.1.2. Climate and Vegetation Factors

NDVI: This study obtained multi-temporal Landsat 8 imagery from the Google Earth Engine platform. Following atmospheric correction and cloud detection, cloud-free or minimally cloudy images were selected to compute the Normalized Difference Vegetation Index [28]. The NDVI formula is as follows:(2)$$\mathrm{NDVI}=\frac{\mathrm{NIR}-\mathrm{Red}}{\mathrm{NIR}+\mathrm{Red}}$$
where NIR represents the reflectance of the near-infrared band and Red represents the reflectance of the red band. This index reflects the influence of various vegetation types and growth conditions on soil stability. Vegetated areas with dense root systems contribute to more stable soil structures, whereas areas with sparse vegetation or withered plants are more susceptible to collapses.

#### 3.1.3. Geological and Human Engineering Activity Factors

Lithology: The rock type, structure, and degree of weathering directly influence slope stability. Lithologies that are prone to weathering or possess well-developed joints are more susceptible to collapses [29].

Distance to Faults: Areas near fault zones experience intense geological activity, resulting in more pronounced rock fragmentation and relatively poorer stability [30].

Distance to Rivers: River systems can erode the base of slopes or weaken the rock-soil mass, affecting the mechanical stability of the slope [31].

Distance to Roads: Road construction, slope excavation, and other human engineering activities can disrupt slope stability, accelerating the occurrence of collapses [32].

Annual Average Rainfall: Long-term meteorological observation data were collected from the National Meteorological Science Data Center. After applying Kriging interpolation, the spatial distribution of annual average rainfall in the study area was derived. This factor is closely related to soil moisture; higher rainfall increases soil moisture content, making slopes more vulnerable to instability [33].

### 3.2. Evaluation of Multicollinearity Among Collapse Evaluation Factors

Multicollinearity poses a significant challenge in hazard modeling as strong correlations among multiple conditioning factors can lead to coefficient inflation and reduced model accuracy [34]. Therefore, it is crucial to identify and quantify multicollinearity before performing hazard modeling. Two common methods for evaluating multicollinearity in input datasets are the variance inflation factor (VIF) and tolerance. The VIF measures the extent to which the variance of the estimated regression coefficients is inflated due to intercorrelations among predictors, while tolerance evaluates the proportion of variance in a given predictor that is not explained by other predictors. VIF values can be calculated as follows:(3)$$VIF_{i}=\frac{1}{1-R_{i}^{2}}$$

In this context, $R_{i}^{2}$ represents the coefficient of determination derived from the regression analysis, where the i-th predictor variable is treated as the dependent variable, and all other predictor variables are used as independent variables.

### 3.3. Collapse Susceptibility Prediction

Collapse susceptibility prediction typically involves analyzing historical collapse events and their associated environmental factors to identify relationships. The first step in predictive modeling is to estimate future collapse probabilities based on historical events and generate a corresponding collapse distribution map. In this study, a total of 651 collapse sites were identified in the study area, based on the 2015 geological survey report. The collapse points were randomly divided into a training set for model development and a testing set for performance evaluation. The training set comprised 456 collapse points (70%), while the testing set consisted of 195 points (30%). This data split ensures reliable model testing and enhances the credibility of the predictions.

Evaluation factor information was extracted from both collapse and non-collapse samples to generate a GIS database. This database, containing information on factors affecting both collapse and non-collapse occurrences, was converted into CSV format for machine learning applications. Based on machine learning model training and testing, the optimal model was selected to predict collapse probabilities for grid points in the study area. The study area was divided into 40,235 grids using a fishnet created in the GIS, with grid sizes of 200 m × 200 m. While collapse susceptibility is typically generated at 30 m resolution, with smaller grid sizes yielding higher accuracy [35], the study area would contain over 17 million grids at 30 m resolution. Previous attempts to calculate collapse probabilities using 30 m grids proved computationally challenging due to the massive data volume. Earlier publications have used grid classifications of 600 m to predict spatial collapse distribution [36]. It should be noted that classification grids are not used to represent collapses themselves. Kriging interpolation analysis was employed to determine probabilities between adjacent points and predict collapse probabilities within specific grids. The susceptibility map was generated through interpolation analysis using center point grids.

Collapse susceptibility modeling is conventionally approached as a binary classification task, where probability values are classified as either indicative of a collapse or not. In this study, the buffer tool in ArcGIS 10.8 was used to delineate circular buffers with a radius of 500 m surrounding each identified collapse point. From these buffers, 651 non-collapse points were randomly selected from areas external to the buffers. The resulting training and testing datasets comprised 911 and 391 non-collapse points, respectively. In the data labeling process, collapse points were assigned a value of 1, while non-collapse points were assigned a value of 0. The methodology for assessing collapse susceptibility is illustrated in Figure 3. Historical locations of collapse events within the study area were catalogued to establish a collapse inventory. The dataset was then used with ten-fold cross-validation to evaluate the model’s performance. A multicollinearity assessment was conducted on the conditioning factors to mitigate potential correlation issues. During the model development phase, hyperparameters of the machine learning model were fine-tuned using particle swarm optimization. Ultimately, the predictive accuracy of collapse susceptibility was assessed using various performance metrics. All machine learning models were implemented in the Python 3.12 programming environment.

#### 3.3.1. Support Vector Machine (SVM)

SVM models are integral to predicting collapse hazards. In particular, the Support Vector Classification (SVC) method demonstrates considerable efficacy in identifying and forecasting potential collapse-prone areas, owing to its advanced classification capabilities. The fundamental principle of SVC involves the establishment of an optimal linear hyperplane, which effectively differentiates between various classes by maximizing the margin between data samples. To address the complexities and nonlinear characteristics inherent in geological data, SVC employs a kernel function. This function enables the SVM to operate in high-dimensional spaces, thereby transforming linear problems into nonlinear ones and facilitating the creation of more adaptable and precise decision boundaries. Such capabilities are essential for managing the nonlinear relationships present in geological datasets. Furthermore, the standardization of input variables is a critical preparatory step in SVM applications. By standardizing the data, the model mitigates the impact of varying units and scales on prediction outcomes, thereby enhancing both model stability and accuracy. This preprocessing phase is vital for improving the model’s generalization capabilities and its applicability across diverse datasets.(4)$$F(x)=\mathrm{sign}\left(\sum_{i=1}^{n}\alpha_{i}y_{i}K(x_{i},x)+b\right)$$

$\alpha_{i}$ is the Lagrange multiplier, denoting the weight of each support vector. $K(x_{i},x)$ is the kernel function that computes the similarity between input sample x and support vector $x_{i}$. b is a bias term to adjust the position of the decision boundary.

#### 3.3.2. Random Forest (RF)

RF models are crucial in collapse hazard prediction. As an ensemble learning technique, Random Forest enhances prediction accuracy and robustness by constructing multiple decision trees. Each decision tree uses a random subset of the input data during training, which not only improves the model’s generalization capability but also effectively reduces the risk of overfitting. During the prediction phase, each tree in the Random Forest makes decisions independently, with the final result determined by a majority vote or average value. This ensemble strategy allows Random Forest to perform effectively when dealing with complex geological data, capturing nonlinear relationships and intricate patterns. The high accuracy and computational efficiency of Random Forest make it an ideal choice for collapse hazard prediction. It is not only capable of handling large-scale datasets, but it also provides feature importance analysis to identify the factors with the greatest impact on collapse risk. This feature provides a scientific basis for the prevention and management of geological hazards, thereby enhancing the reliability and applicability of predictions [37].(5)$$\hat{y}=\frac{1}{B}\sum_{b=1}^{B}f_{b}(x)$$
where $\hat{y}$ is the final prediction, which is the average of all tree predictions, and B is the number of total decision trees. The prediction result of the bth tree for input sample x is calculated.

#### 3.3.3. Extreme Gradient Boosting (XGBoost)

In predicting collapse hazard, XGBoost has garnered significant attention due to its superior performance in addressing complex geological and environmental factors [38]. The model employs an iterative approach to construct decision trees, with each subsequent tree being developed based on the residuals of its predecessor. This methodology enables each iteration to focus on the data segments where prior trees exhibited deficiencies, thereby progressively enhancing the model’s predictive capabilities. The ultimate predictions regarding collapse hazard are derived by aggregating the outputs of these trees, and this ensemble technique significantly improves the model’s accuracy. Furthermore, XGBoost effectively mitigates the risk of overfitting through regularization, ensuring robust performance on both training datasets and previously unseen data. In collapse hazard prediction, the objective function is designed to enhance generalization through regularization, with the algorithm’s goal being the minimization of this objective function to optimize model performance. Consequently, XGBoost emerges as a formidable tool in geohazard prediction, yielding reliable predictive outcomes and comprehensive characterizations.(6)$$F=\sum_{i=1}^{N}L(y_{i},{\hat{y}}_{i})+\sum_{j}\Omega (f_{j})$$

$L(y_{i},{\hat{y}}_{i})$ is used to measure the difference between the actual collapse state $y_{i}$ and the predicted value ${\hat{y}}_{i}$. $\sum_{j}\Omega (f_{j})$ is a regularization term that smooths out the final learning weights and prevents overfitting. In both equations, i denotes the number of samples in the dataset and N is the total amount of data imported into the jth tree.(7)$$\Omega (f_{j})=\gamma T+\frac{1}{2}\lambda w^{2}$$

T is the number of leaf nodes in the tree and w denotes the fraction of each leaf. γ and λ are regularization parameters used to adjust the tree complexity. In minimizing the objective function, an iterative approach is used. The objective function to be minimized in step t is shown in the following equation:(8)$$F^{\prime }=\sum_{i=1}^{N}L(y_{i},{\hat{y}}_{i}^{t-1}+f_{t}(x_{i}))+\Omega (f_{t})$$

This iterative optimization mechanism of the XGBoost model in predicting collapse hazard enables it to effectively capture complex nonlinear relationships [39] between topographic factors, hydrological factors, and geological factors. Through eigen-sampling and regularization, the model reduces the risk of overfitting and improves accuracy in areas not predicted by prior models.

#### 3.3.4. Light Gradient Boosting Machine (LightGBM)

LightGBM is an efficient framework for gradient boosting decision trees developed by Microsoft. It enhances training speed and memory efficiency by employing histogram-based algorithmic optimization and a leaf-first growth strategy [40]. In predicting collapse hazard, LightGBM exhibits notable advantages due to its superior feature processing capabilities and effective computational performance [41].(9)$$L=\sum_{i=1}^{n}l\left(y_{i},\hat{y_{i}}\right)+\sum_{k=1}^{K}\Omega \left(f_{k}\right)$$

In LightGBM, the weights of the leaf nodes determine the contribution of each feature to the prediction result.(10)$$w_{j}=-\frac{\sum_{i\in I_{j}}g_{i}}{\sum_{i\in I_{j}}h_{i}+\lambda }$$

$w_{j}$ is the weight, $I_{j}$ is the sample set sample of the jth leaf node, $g_{i}$ is the first-order derivative of sample i, which represents the gradient of the loss function with respect to the predicted value, and $h_{i}$ the second-order derivative of sample i, which represents the curvature of the loss function. λ is a regularization parameter that controls the model complexity and prevents overfitting.

### 3.4. Model Accuracy Evaluation

To evaluate the model’s predictive performance in collapse hazard prediction, we employed a range of metrics: precision, accuracy, recall, F1-score, the receiver operating characteristic (ROC) curve, and the area under the curve (AUC). These metrics provide a comprehensive assessment of the model’s effectiveness in identifying susceptible areas.(11)$$Precision=\frac{TP}{TP+FP}$$

Precision quantifies the accuracy of positive predictions and is defined as the ratio of true positives (TPs) to the sum of true positives and false positives (FPs). This metric reflects the model’s proficiency in accurately identifying collapse events within the set of all predicted collapses. In this context, TP refers to the number of samples accurately classified as collapse events, whereas FP refers to the samples erroneously predicted as collapse events.(12)$$Accuracy=\frac{TP+TN}{TP+TN+FP+FN}$$

Accuracy serves as a measure of a model’s overall correctness in predictions. It is quantitatively expressed as the ratio of correctly predicted instances, including both true positives (TPs) and true negatives (TNs), to the total number of instances evaluated. This metric accounts for true negatives (TNs), which represent the samples correctly classified as non-collapse events, and false negatives (FNs), which denote the samples erroneously classified as non-collapse events. Recall, also known as sensitivity, assesses the model’s efficacy in detecting actual collapse events. It is quantified as the ratio of true positives to the sum of true positives and false negatives. This metric reflects the model’s ability to identify all relevant instances.(13)$$Recall=\frac{TP}{TP+FN}$$

The F1-score represents the harmonic mean of precision and recall, offering a comprehensive evaluation of a model’s accuracy and completeness. This metric is particularly advantageous in the context of imbalanced datasets as it considers both false positives and false negatives.(14)$$F1-score=2\times \frac{Precision\times Recall}{Precision+Recall}$$

### 3.5. Interpretability of Machine Learning Models

Beyond emphasizing the significance of various features, a more in-depth investigation into the interpretability of machine learning models was conducted through the analysis of SHAP values [42]. These values provide a robust framework for understanding predictions made by machine learning algorithms as they quantify the extent to which each feature influences predictions for specific instances [43]. This analysis clarifies the impact of SHAP values on the features defined by each machine learning algorithm [44].(15)$$SHAPvalue=\sum_{S\subseteq N\setminus \{i\}}\frac{|S|!(|N|-|S|-1)!}{|N|!}\left[f(S\cup \{i\})-f(S)\right]\;$$
where SHAPvalue represents the contribution of the ith feature in a single collapse sample. N the set of all features in the training set. S is the subset that does not contain i features. f(S∪{i}) represents the model output when feature i is added to subset S. The weight $\frac{|S|!(|N|-|S|-1)!}{|N|!}$ reflects the importance of the subset in the Shapley value calculation.

According to Shapely’s theory which follows the additivity of features, the output of a single sample feature can be defined as follows:(16)$$f\left(x\right)=h\left(z^{\prime }\right)=\phi_{0}+\sum_{i=1}^{N}\phi_{i}z_{i}^{\prime }$$

f(x) denotes the predicted value of the model for the current sample. $h(z^{\prime })$ is the sum of the Shapley values corresponding to all features, indicating the total contribution of the features to the predicted values of the model. $\phi_{0}$ represents the average predicted value across all samples and is often considered the baseline value for the model. $h(z^{\prime })$ indicates whether the contribution of the ith feature is included in the current feature subset, with a value of 1 if the feature is included and 0 if not.

### 3.6. Hazard Assessment

Hazard is the primary element in collapse geohazards. Hazard assessment primarily involves two aspects: the spatial probability and temporal probability of disaster occurrence. Spatial probability refers to the likelihood of an occurrence at specific locations under the condition of an induced event, i.e., susceptibility; temporal probability refers to the frequency or intensity of the induced factor [45]. According to the hazard definition, for the triggering factor of volcanic earthquakes, the product of susceptibility and the probability of exceeding the threshold of volcanic earthquakes is used to model hazard assessment.(17)H=P(s)×P(t)
where P(s) represents the spatial probability of collapse occurrence, i.e., susceptibility, and P(t) denotes the temporal probability.

To assess the impact of Changbai volcanic earthquakes on collapse geohazards in the study area, this research study utilizes the likelihood of volcanic earthquakes with an intensity exceeding level VI to evaluate their influence on geotechnical stability in the region (26). The detailed calculation process is outlined as follows:

(1) The Gutenberg–Richter (G-R) law describes the relationship between earthquake magnitude and the cumulative frequency of earthquakes, which is commonly referred to as the G-R relationship. The G-R relationship, one of the fundamental laws of seismology, has been extensively applied in studies of seismicity, earthquake prediction, and seismic zoning. The formula is as follows:(18)lgN(M)=−aM+b
where M represents the magnitude, N(M) denotes the cumulative frequency of earthquakes, and a and b are constants.

(2) Currently, limited research exists on the relationship between the VEL value of the volcanic eruption index and the maximum magnitude of the associated earthquake. Referring to the 1991 eruption of the Pinatubo volcano in the Philippines, which had an eruption index of 6, the maximum magnitude of the accompanying earthquake was 5.8. The relationship between earthquake magnitude, the volcanic eruption index, and energy is characterized by exponential relationships. Under normal circumstances, the higher the eruption index, the greater the corresponding increase in the magnitude of the induced earthquake. This study assumes that the maximum magnitude of associated earthquakes is equal to the eruption index, thus establishing the following relationship:(19)lgN(VEL)=−0.2286M+1.6286
where M is the magnitude of earthquakes associated with volcanic activity in the Changbai Mountain area, and N(VEL) is the volcanic eruption index.

(3) Numerous experts and scholars, both domestically and internationally, have conducted statistical analyses on the relationship between seismic intensity and collapse disasters, concluding that there is a direct correlation between the distribution of earthquake-triggered collapses and earthquake intensity. Specifically, when the seismic intensity reaches or exceeds a VI-degree zone, the number of earthquake-triggered collapses and collapse disasters increases significantly. Therefore, the seismic intensity attenuation formula for the study area can be expressed as follows:(20)$$L=1.454M_{S}-1.792\ln(R+16)+4.493$$
where L is the seismic intensity, $M_{S}$ is the magnitude of an earthquake when it occurs, and R is the epicenter distance. In this paper, the epicenter is selected as the location of Tianchi Mouth, with coordinates 42.0035° N, 128.0538° E.

(4) During seismic energy propagation, the energy gradually attenuates, and the seismic intensity decreases progressively with increasing epicenter distance [46]. The probability of seismic intensity exceeding level VI, generated by the Changbaishan volcanic earthquake, is expressed in the following formula for the 5000-year study time span:(21)$$P_{E}=\frac{\int_{k}^{9}\frac{1}{5000}\times 10^{-0.2286M+1.6286}}{\int_{0}^{9}\frac{1}{5000}\times 10^{-0.2286M+1.6286}}$$
where k denotes the required earthquake magnitude at seismic intensity VI caused by the Changbai volcano earthquakes in the study area, and P_E_ is the probability of seismic intensity exceeding VI.

In the ArcGIS platform, using Tianchi as the epicenter of the volcano-associated earthquake, the zoning map of the study area for the probability of exceeding is generated according to the probability of exceeding P_v_ generated by different epicentral distances R. The following table shows the probability of exceeding in the study area for different magnitudes of earthquakes.

## 4. Results

### 4.1. Analysis of Parameter

This study integrates multiple dimensional factors, including topography, climate, vegetation, and human engineering activities, utilizing remote sensing data and machine learning methods to comprehensively assess collapse hazard in the Changbai Mountain region. The combination of topographic and climate–vegetation factors effectively quantifies the comprehensive impact of topography on moisture convergence and slope stability. An increase in the annual average rainfall significantly enhances soil moisture, thereby increasing the instability of soil at potential collapse points. The application of the NDVI further underscores the critical role of vegetation cover in soil stabilization. Furthermore, the comprehensive consideration of geological factors and human engineering activities fully reflects the combined impact of natural conditions and human activities on collapse risk. This study indicates that slopes near roads are more prone to instability due to excavation and construction activities, while areas near rivers are relatively more stable due to sufficient moisture and higher soil humidity. The visualization is shown in Figure 4.

Through the acquisition of remote sensing data, processing using ArcGIS 10.8 and ENVI 5.6, and the integration of machine learning models, this research studyprovides a systematic method for assessing collapse hazard. The findings provide scientific evidence for disaster prevention and risk management in the Changbai Mountain region and similar geological environments.

### 4.2. Multicollinearity Analysis

The role of feature factors in the performance of machine learning models is pivotal as they directly influence the accuracy and reliability of prediction outcomes. Accurately identifying and selecting robust influencing factors is a critical step in achieving desirable prediction outcomes. To support the selected influencing factors, this study reviews previous studies on collapse hazard evaluation and analyzes the potential issue of multicollinearity. Typically, a VIF greater than 10 or a tolerance value below 0.1 indicates a significant level of multicollinearity, necessitating further investigation and potential correction [47]. The VIF and tolerance values for each factor are presented in Table 2, providing a comprehensive view of the multicollinearity analysis results.

In Figure 5, the relationship between each pair of influencing factors was assessed through the Pearson correlation analysis. Correlation coefficients nearing 1 indicate a strong positive relationship, while coefficients approaching 0 suggest independence between variables. Previous research suggests that correlation coefficients exceeding 0.8 may cause multicollinearity issues, which could compromise model stability. Based on the analysis, the factor TWI, which exhibited a correlation above 0.8 with other variables, was removed from the modeling process to ensure the accuracy and reliability of the results. The remaining influencing factors showed correlation values below 0.8, indicating that they operate independently and are suitable for use as input variables in the model.

### 4.3. Tuning of Hyperparameters

The optimization of hyperparameters, which function as critical regulatory variables in machine learning models, significantly impacts model performance. This study utilizes the particle swarm optimization (PSO) algorithm for the automated tuning of parameters to enhance the model’s predictive performance. Initially, the input variables are standardized, followed by an evaluation of model performance using cross-validation. The PSO algorithm demonstrates a distinct advantage in ensemble learning methods, such as Random Forests and boosting, as it optimizes the parameters of individual learners while also fine-tuning the overall architecture of the ensemble model. This dual optimization improves the model’s ability to generalize to previously unseen data. The optimal combinations of hyperparameters identified by the PSO algorithm are presented in the following Table 3.

### 4.4. Collapse Susceptibility Assessment

The prediction outcomes were visualized using the ArcGIS 10.8 platform, with Figure 6 and Table 4 presenting the collapse hazard prediction results alongside the reclassified frequency ratios for the RF, SVM, XGBoost, and LightGBM models, classified into five categories: very low, low, medium, high, and very high. The findings indicate that areas with medium to high hazard are primarily located along both sides of roadways and in regions characterized by significant human activity. In terms of model predictive performance, the results from the RF and SVM models exhibit a high degree of similarity, with high-hazard zones identified primarily along National Highway G301, in Wanliang Town, and near Mingyue Town within Changbai County. This consistency suggests that both models are reliable in identifying high-risk zones. Conversely, the predictions generated by the XGBoost model tend to favor regions with lower vegetation cover, particularly near Fusong County.

### 4.5. Model Evaluation

The SVM model demonstrates comprehensive performance across all assessed metrics, achieving an accuracy of 0.9094, a precision of 0.9068, a recall of 0.8992, and an F1-score of 0.9030, as shown in Table 4. Furthermore, it achieved a notable area under the curve–receiver operating characteristic (AUC-ROC) score of 0.941, highlighting its robust overall performance and proficiency in sample identification. In contrast, the Random Forest model excelled in the AUC-ROC metric, attaining an impressive score of 0.968, thereby demonstrating its reliability in classification and strong performance. It also recorded a recall of 0.9050 and an F1-score of 0.9005, indicating its effectiveness in balancing the identification of both positive and negative samples. The XGBoost model is distinguished by a notable recall rate of 0.9347, emphasizing its enhanced ability to detect collapsed samples. It also boasts an F1-score of 0.9051 and an AUC-ROC of 0.955, further reinforcing its strong overall performance. Conversely, the LightGBM model leads with a precision score of 0.9037, reflecting its accuracy in predicting collapse events. Additionally, it achieves an AUC-ROC of 0.960 and an F1-score of 0.9004, demonstrating its commendable overall effectiveness.

A comprehensive analysis demonstrates that each model performs well in the task of collapse hazard prediction. The Random Forest model stands out in key indicators, particularly the AUC-ROC, making it the overall optimal model. XGBoost and SVM each have their unique characteristics, making them suitable for specific application scenarios. LightGBM, with its distinctive advantage in accuracy, is well suited for scenarios that require high prediction precision. These findings provide strong technical support for future collapse hazard research. See Figure 7.

Overall, the XGBoost model excels in identifying and differentiating high-risk regions, demonstrating superior frequency ratios and prediction accuracy compared to the other models. These results emphasize the XGBoost model’s exceptional overall predictive performance and its capability in high-risk identification. Consequently, this study selects the XGBoost model for detailed explanatory analysis to uncover the intrinsic properties of its prediction mechanism, offering valuable insights for the risk assessment, prevention, and management of collapse disasters.

### 4.6. Hazard Assessment

Using Changbaishan Tianchi as the epicenter and defining various epicenter distances, the probability of volcanic earthquakes exceeding a certain threshold was calculated in ArcGIS 10.8. The raster calculator was then employed to overlay the results with the hazard data, generating the collapse hazard zoning map for the Changbaishan area. See Figure 8.

Figure 9 presents the distribution of raster proportions across four models at varying hazard levels. The XGBoost model predicts the following proportions: very low (35.46%), low (29.65%), medium (18.37%), high (11.17%), and very high (5.33%). In comparison, the Random Forest (RF) model produces the following proportions: very low (33.93%), low (29.38%), medium (18.61%), high (13.77%), and very high (4.31%). The Support Vector Machine (SVM) model yields the following proportions: very low (26.53%), low (25.97%), medium (20.40%), high (14.74%), and very high (12.33%). These results suggest that both the XGBoost and RF models demonstrate a higher sensitivity in predicting low-risk areas while adopting a more conservative approach for high-risk regions. 

The ability of the models to distinguish between different hazard categories was further assessed by calculating the ratio of the total frequency ratio for the high- and very-high-hazard zones relative to the sum of the total frequency ratios. As shown in Table 5, the frequency ratios for all four models were below 1 in the very-low-, low-, and medium-hazard zones, indicating strong predictive performance. In contrast, the Random Forest (RF) model achieved frequency ratios of 2.773 and 8.345, with an accuracy of 0.903 in the high- and very-high-hazard zones, respectively. The Support Vector Machine (SVM) model produced frequency ratios of 1.605 and 3.061, with an accuracy of 0.734, while the XGBoost model had frequency ratios of 1.935 and 12.954, respectively.

### 4.7. Variable Importance

In the present study, the marginal effect methodology outlined by SHAP was employed to conduct a comprehensive analysis of the contribution of each evaluative factor within the XGBoost model concerning the incidence of collapse. As shown in Figure 10, the color bars represent the actual values of each factor, while the horizontal axis illustrates the distribution of SHAP values. A positive SHAP value indicates that a particular factor has a beneficial effect on the likelihood of collapses, whereas a negative value suggests a detrimental impact. Specifically, elevation, the NDVI, and distance to rivers and roads are negatively correlated with collapse incidence. This suggests that these factors may reduce the risk of collapse under specific conditions or ranges. Elevation may reduce collapse risk by influencing water flow patterns and soil stability, while the NDVI may influence this through the stabilizing effect of vegetation cover. Distance to rivers and roads may reflect the effects of topography and human activities. In contrast, faulting and stratigraphic lithology are positively correlated with collapse probability. This is likely due to the fact that fault activity increases geological instability, while specific lithologies are more susceptible to weathering and disintegration, thus increasing the risk of collapse.

Through the analysis of SHAP values, we quantified and visualized the influence of each factor, thereby providing a transparent understanding of the model’s decision-making process. This approach not only reveals the relative importance of each factor but also helps identify potential nonlinear relationships and interaction effects, thus providing a scientific basis for collapse risk management.

Areas within 0–10,000 m from the fault exhibit higher positive SHAP values, and as the distance increases (>20,000 m), the impact gradually stabilizes and slightly increases, suggesting that proximity to the fault increases collapse susceptibility, likely due to tectonic activity and geological fragmentation zones. The slope gradient exhibits a clear positive correlation with collapse incidence. When the slope gradient is between 0 and 10 degrees, SHAP values rise rapidly, and the impact continues to increase gradually beyond 20 degrees, suggesting that slope gradient is a key factor influencing collapse occurrence, with steeper slopes being more susceptible. NDVI values exhibit a negative correlation when ranging from 0.3 to 0.8, with SHAP values significantly decreasing as the NDVI approaches 1 (high vegetation cover), indicating that greater vegetation cover reduces collapse risk. Within 0–2000 m from the road, SHAP values decrease sharply and level off beyond 2000 m, indicating that human activities (such as road construction) significantly influence collapse risk in the proximal zone. Higher positive SHAP values are observed at low elevations (<500 m), with the effect gradually decreasing and becoming negative as the elevation increases, likely due to topographic features and climatic conditions. Positive impacts were observed in the range of 7000–8000 mm, with relatively low impacts at both very low and very high rainfall, suggesting that moderate rainfall increases collapse risk. Slope orientation has a relatively minor effect on collapses, with minimal fluctuation in SHAP values and a nearly horizontal curve, indicating that it is not a determining factor. Significant variation in impacts was observed near 0 m from the river, with stabilization after moving beyond 2000 m, suggesting that fluvial erosion significantly influences collapse risk in the immediate area. The range from −2 to 2 exhibits a significant U-shaped relationship, with extreme curvature values (both positive and negative) showing higher SHAP values, indicating that concave and convex terrain variations significantly influence collapse occurrence. Different lithological types exhibit smaller variations in SHAP values, with a slight overall increase, suggesting that lithology has a relatively minor effect on collapses.

The results, as illustrated in Figure 11, indicate that collapse occurrence is influenced by a combination of environmental factors, with slope, vegetation cover, and proximity to roads identified as the most significant contributors. These findings offer valuable insights into the spatial distribution of collapse-prone areas, providing a scientific basis for risk assessment and the development of effective preventive measures.

### 4.8. Interaction of Distance from Road and NDVI

Figure 12 illustrates the interaction between road distance and the NDVI in relation to collapse hazard. In the proximal zone (0–2000 m), SHAP values exhibit significant fluctuations (−0.3 to 0.5), showing maximum sensitivity to NDVI variations, indicating the critical zone for collapse hazard. The transitional zone (2000–4000 m) shows a convergence of SHAP values to a narrower range (−0.15 to −0.1), indicating attenuated road disturbance and moderated NDVI effects. In the distal zone (>4000 m), SHAP values stabilize with significantly reduced fluctuations, suggesting substantially diminished road influence.

The regulatory effects of the NDVI on collapse hazard occur in three distinct intervals: when NDVI < 0, SHAP values show a significant negative correlation, indicating increased collapse risk due to insufficient vegetation cover; within the range of 0 to 0.3, SHAP values exhibit maximum volatility, representing a critical transition zone; and when NDVI > 0.3, SHAP values stabilize, suggesting reduced marginal effects. In the proximal zone, high NDVI values (warm colors) correspond to smaller negative SHAP values, suggesting that robust vegetation cover can mitigate road-induced effects, while low NDVI values (cool colors) correspond to larger fluctuations in SHAP values, indicating that areas with sparse vegetation are more susceptible to road disturbance.

The analysis reveals a critical threshold distance of approximately 2000 m, beyond which road influence significantly decreases, and identifies the NDVI range of 0–0.3 as exhibiting the strongest regulatory effect, providing crucial guidance for vegetation restoration targets. The comprehensive findings demonstrate that the interaction between road distance and the NDVI significantly influences collapse hazard, emphasizing the priority of vegetation restoration and protection measures in road-adjacent areas. These insights provide a valuable reference for collapse risk assessment and mitigation strategies.

## 5. Discussion

The main contribution lies in the integration of machine learning techniques with SHAP analysis for collapse hazard assessment, with PSO improving the predictive accuracy of the models. This approach has resulted in the development of reliable collapse hazard maps, which offer substantial support for disaster management [48]. However, several aspects still require further exploration and refinement.

The integration of collapse hazard maps into disaster management systems has been explored but remains underdeveloped [49]. Future research should focus on incorporating the risk assessment results from this study into operational disaster management systems. For instance, these hazard maps could be incorporated into real-time disaster monitoring and early warning systems to support managers in making informed decisions before, during, and after disasters. Through integration with GIS platforms or other disaster management tools, policymakers can prioritize resource allocation and implement effective preventive measures in high-risk areas. Further work will continue to explore methods to achieve this integration and provide practical operational solutions for disaster management.

Although the advantages of machine learning and SHAP analysis are demonstrated, a comparison with neural networks or hybrid methods is absent. Neural networks and hybrid approaches have considerable potential in addressing complex nonlinear relationships and could further improve the model’s predictive performance [50]. Future studies may compare these advanced methods with the current models, assessing their performance in collapse hazard assessment and exploring whether they offer improved accuracy or better generalization.

The impact of climate change and human activities is a significant factor not addressed in the analysis. Climate change may alter precipitation patterns, thus affecting the frequency and severity of collapses, while human activities—such as urbanization, land development, and infrastructure construction—directly or indirectly modify regional collapse risks. Although basic environmental factors were considered during data collection, temporal variations were not incorporated. Future research should integrate long-term data on climate change and human activities to assess their potential impact on collapse hazards. By incorporating time series analysis [51], models can more accurately capture the dynamic changes in these factors, thereby improving the accuracy and timeliness of predictions.

The findings are of significant importance to the Changbai Mountain region, and the methodology and results can be applied to other regions with similar geological and climatic conditions. The integration of machine learning and SHAP analysis provides robust adaptability, enabling adjustments to be made based on varying geographical and environmental conditions. Future research could apply this approach to other regions, further validating the universality and effectiveness of the method while comparing its results to those from other regions to enhance its global applicability.

A robust framework for collapse hazard assessment is established, and future research will incorporate additional variables, utilize more advanced algorithms, and account for the dynamic impacts of climate change and human activities. These efforts will enhance the accuracy of collapse hazard assessments and provide more substantial support for disaster management and mitigation decision making.

## 6. Conclusions

This study highlights the significant threat of collapse disasters to societal safety and economic stability in the Changbai Mountain region, characterized by complex geological and climatic conditions. By employing PSO to optimize four commonly used machine learning models, we were able to incorporate topographic, climatic, and anthropogenic factors to predict collapse hazards and generate detailed hazard maps. The strong predictive capabilities of these models, evidenced by AUC values exceeding 0.9, underline their robustness in identifying collapse-prone areas, making them invaluable for disaster preparedness and risk mitigation. The application of SHAP theory provided a deeper understanding of key risk factors such as proximity to roads, the NDVI, and slope, offering actionable insights to guide mitigation strategies.

Beyond the Changbai Mountain region, the methods developed in this study, integrating machine learning with SHAP analysis, are transferable to other regions with similar geological hazards. This framework serves as a foundation for future research in geohazards, enhancing our understanding of environmental interactions and improving risk prediction. While the results are promising, further research should incorporate time series data to account for climate change and human activities, as well as higher-resolution models to refine hazard maps. Ultimately, this study contributes to the development of proactive disaster risk management strategies, safeguarding lives, infrastructure, and ecosystems.

## Figures and Tables

**Figure 1 sensors-25-01512-f001:**
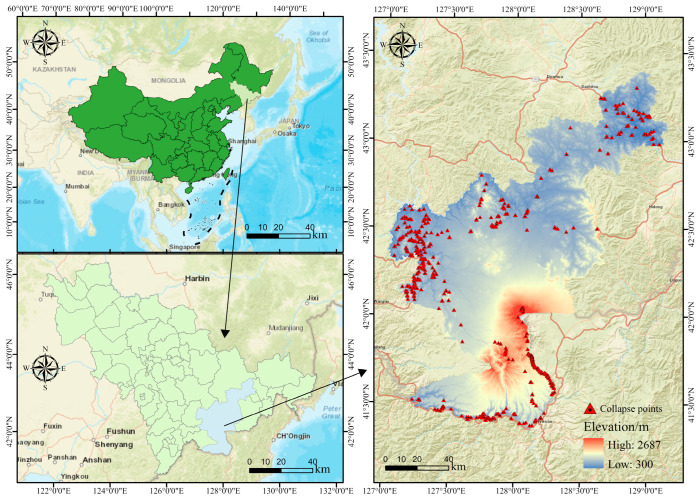
Spatial distribution of collapse sites within the study area.

**Figure 2 sensors-25-01512-f002:**
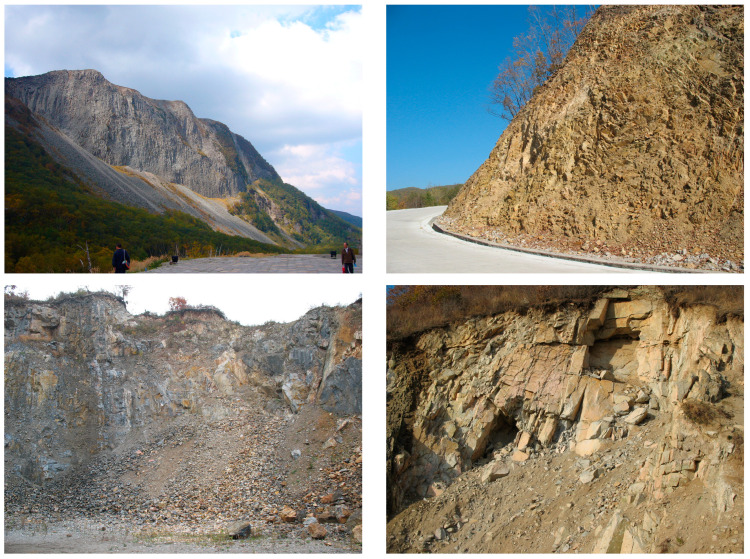
Field photos of collapse geological disaster sites in the study area.

**Figure 3 sensors-25-01512-f003:**
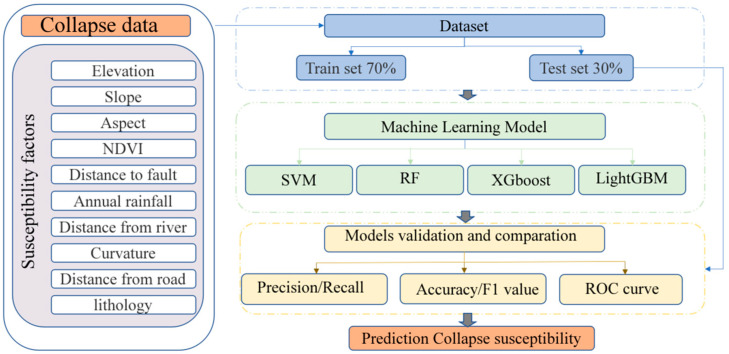
Flowchart for prediction of collapse.

**Figure 4 sensors-25-01512-f004:**
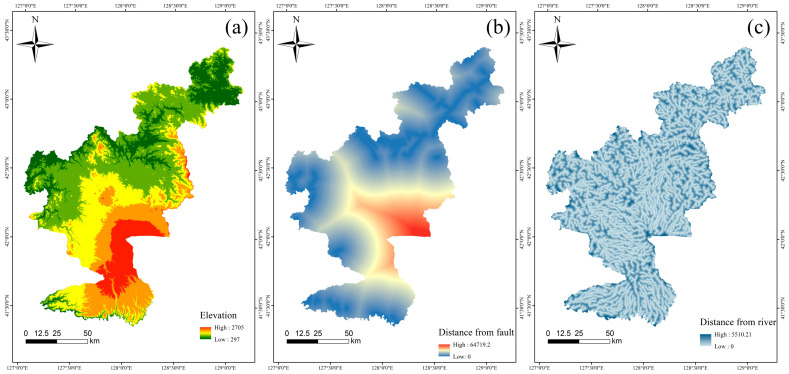
(**a**) Elevation; (**b**) distance from faults; (**c**) distance from river; (**d**) aspect; (**e**) distance from road; (**f**) NDVI; (**g**) annual average rainfall; (**h**) slope; (**i**) lithology; (**j**) curvature; (**k**) TWI; (**l**) exceedance probability.

**Figure 5 sensors-25-01512-f005:**
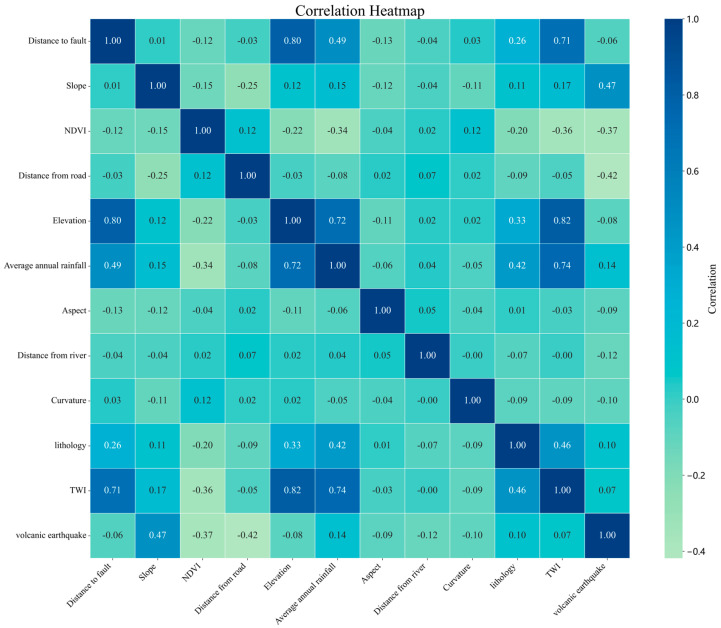
Pearson’s correlation coefficient.

**Figure 6 sensors-25-01512-f006:**
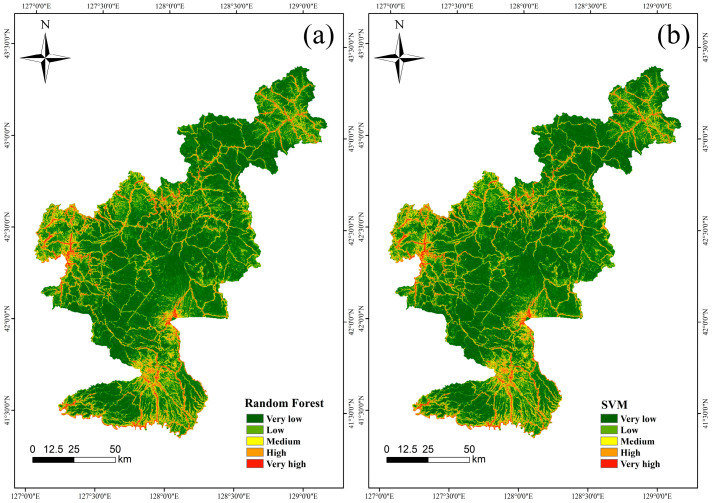
Collapse susceptibility mapping: (**a**) Random Forest; (**b**) SVM; (**c**) XGBoost; (**d**) LightGBM.

**Figure 7 sensors-25-01512-f007:**
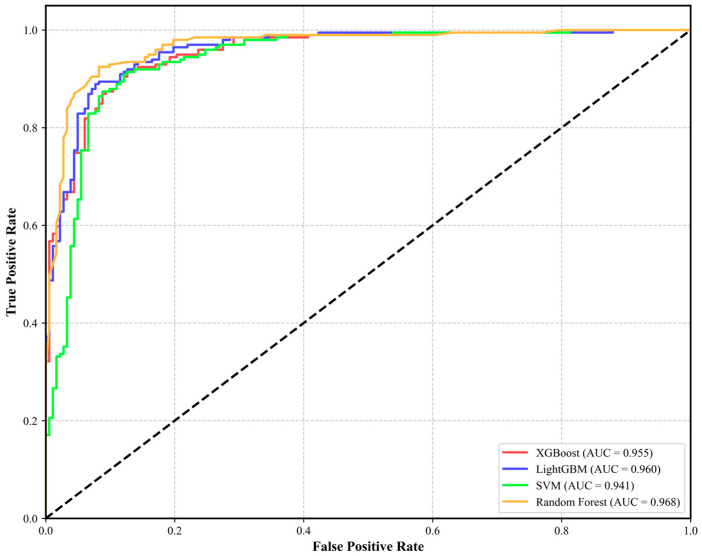
ROC curves for the four models and the AUC values.

**Figure 8 sensors-25-01512-f008:**
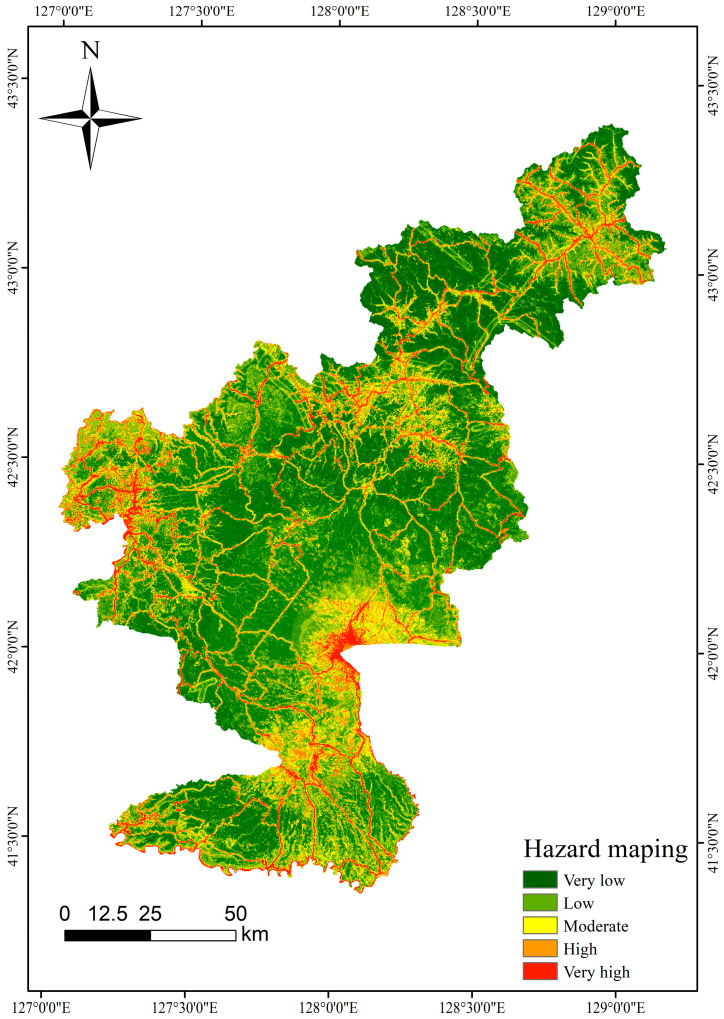
Hazard zoning.

**Figure 9 sensors-25-01512-f009:**
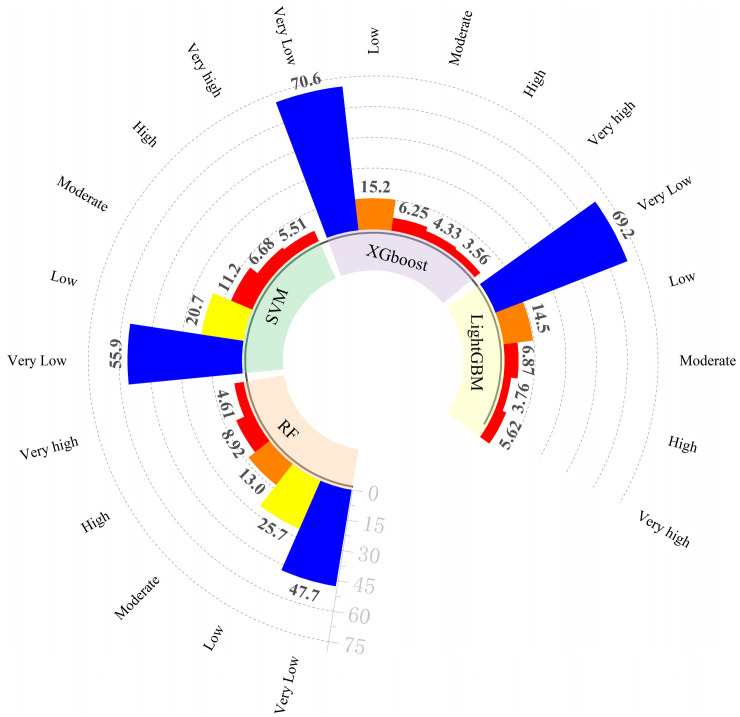
Hazard subdivision and raster occupancy map.

**Figure 10 sensors-25-01512-f010:**
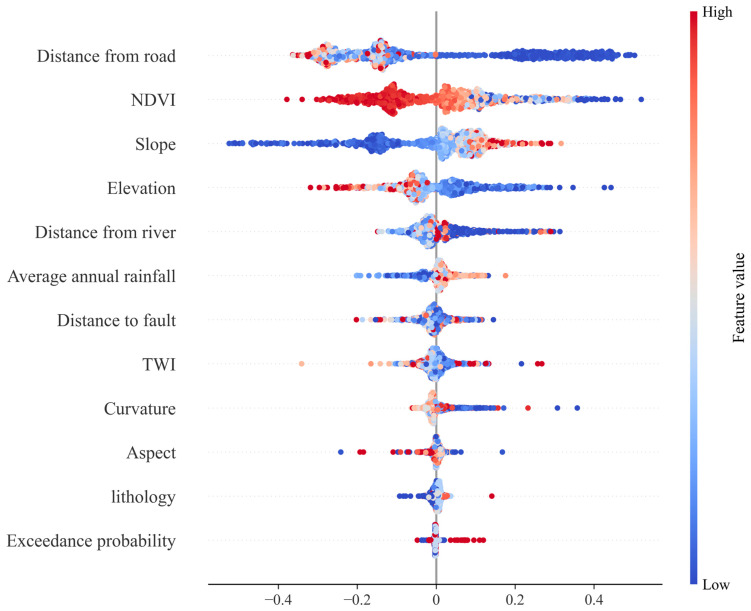
SHAP summary map.

**Figure 11 sensors-25-01512-f011:**
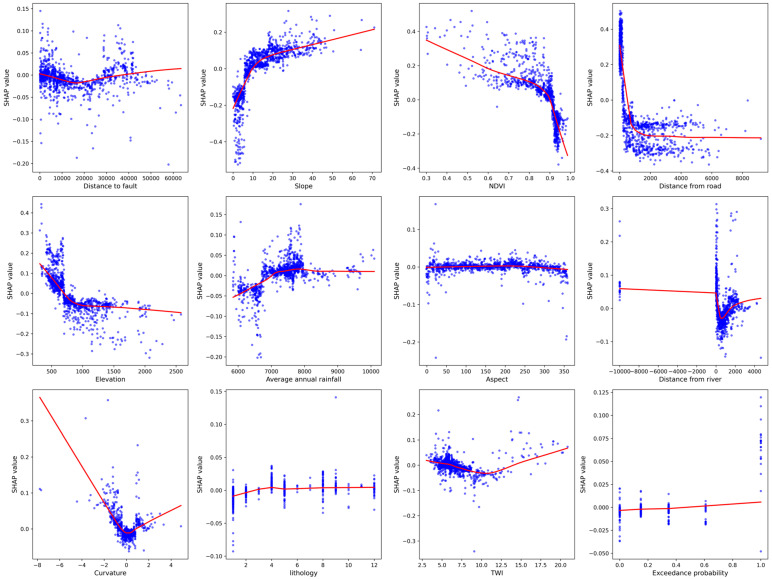
Shap dependency diagram.

**Figure 12 sensors-25-01512-f012:**
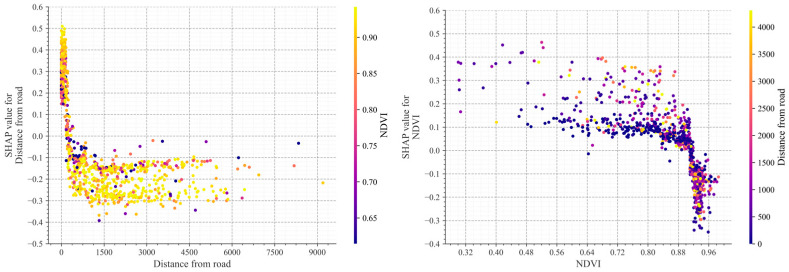
Shap explains the extent to which the collapse contributes to the distance from the road and the NDVI.

**Table 1 sensors-25-01512-t001:** Summary of the main data sources.

Indicator	Data Type	Data Source
DEM	Raster	ASTGTM2 DEM (30 m), GIS analysis
Slope, aspect, curvature	Raster	Derived from DEM
Topographic wetness index	Raster	ASTGTM2 DEM (30 m), GIS analysis
Surface roughness	Raster	ASTGTM2 DEM (30 m), GIS analysis
Lithology	Vector	National Geological Data Center (http://dc.ngac.org.cn/Home accessed on 26 February 2025)
Distance to faults	Vector	National Geological Data Center or relevant geological databases
Distance to rivers	Vector	National Geographic Information Resource Catalog Service System (https://www.webmap.cn accessed on 26 February 2025)
Distance to roads	Vector	National Geographic Information Resource Catalog Service System (https://www.webmap.cn accessed on 26 February 2025)
NDVI	Raster	Monthly average Landsat 8/Sentinel-2 remote sensing data extracted using ENVI5.6
Average annual rainfall	Raster	Average annual rainfall from 2010 to 2022 extracted from the National Meteorological Science Data Center (http://data.cma.cn accessed on 26 February 2025)

**Table 2 sensors-25-01512-t002:** VIF and tolerance.

Feature	VIF	Tolerance
Distance to fault	2.973643075	0.336287838
Slope	1.15067503	0.869055097
NDVI	1.173034736	0.852489674
Distance from road	1.088949798	0.918315979
Elevation	4.58418887	0.218141099
Average annual rainfall	2.495910971	0.400655316
Aspect	1.046434297	0.955626171
Distance from river	1.028486157	0.972302829
Curvature	1.039965286	0.961570557
Lithology	1.256369657	0.795944087
TWI	1.423558	0.702465
Exceedance probability	4.521353	0.221173

**Table 3 sensors-25-01512-t003:** Hyperparameters.

Algorithm	Hyperparameters	Definition	Value
XGboost	n_estimators	Boosting tree count	500
	max_depth	Tree depth limit	6
	learning_rate	Step size	0.1
	subsample	Sample fraction	0.8
	colsample_bytree	Feature fraction	0.8
	gamma	Minimum loss reduction	0.1
LightGBM	n_estimators	Boosting iterations	100
	learning_rate	Step size	0.1
	max_depth	Tree depth limit	5
	num_leaves	Max leaves per tree	31
SVM	C	Regularization parameter	10
	gamma	RBF kernel coefficient	0.1
	kernel	Kernel type	
Random Forest	n_estimators	Forest tree count	300
	max_depth	Tree depth limit	30
	min_samples_split	Min samples for split	2
	min_samples_leaf	Min samples at leaf	1

**Table 4 sensors-25-01512-t004:** Model performance evaluation.

Model	Accuracy	Precision	Recall	F1-Score	AUC-ROC
SVM	0.9094	0.9068	0.8992	0.9030	0.941
RF	0.8974	0.8960	0.9050	0.9005	0.968
XGB	0.8976	0.8774	0.9347	0.9051	0.955
LGBM	0.8962	0.9037	0.8971	0.9004	0.960

**Table 5 sensors-25-01512-t005:** Hazard levels and frequency ratio.

Model	Classification	Number of Grids	Percentage (%)	Number of Collapses	Percentage (%)	Frequency Ratio
RF	1	191,134	47.72%	8	1.28%	0.03
	2	102,975	25.71%	42	6.72%	0.26
	3	52,200	13.03%	38	6.08%	0.47
	4	35,720	8.92%	107	17.12%	1.92
	5	18,469	4.61%	430	68.80%	14.92
SVM	1	223,868	55.90%	13	2.08%	0.04
	2	82,788	20.67%	51	8.16%	0.39
	3	45,008	11.24%	44	7.04%	0.63
	4	26,768	6.68%	80	12.80%	1.92
	5	22,057	5.51%	437	69.92%	12.69
XGBoost	1	282,916	70.64%	35	5.60%	0.08
	2	60,927	15.21%	67	10.72%	0.70
	3	25,042	6.25%	76	12.16%	1.95
	4	17,330	4.33%	114	18.24%	4.21
	5	14,274	3.56%	333	53.28%	14.97
LightGBM	1	277,283	69.24%	34	5.44%	0.08
	2	58,111	14.51%	66	10.56%	0.73
	3	27,498	6.87%	60	9.60%	1.40
	4	15,075	3.76%	74	11.84%	3.15
	5	22,522	5.62%	391	62.56%	11.13

## Data Availability

The raw data supporting the conclusions of this article will be made available by the authors on request.

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
