# Peer review of "Interpretable Machine Learning for Explaining and Predicting Collapse Hazards in the Changbai Mountain Region"

_sensors, 2025, doi:10.3390/s25051512_

Round 1

Reviewer 1 Report

Comments and Suggestions for Authors

Review Comments

Manuscript ID: sensors-3448839

Title: Interpretable Machine Learning for Explaining and Predicting Collapse Hazards in the Changbai Mountain Region

This manuscript presents a significant contribution to the application of interpretable machine learning for assessing and predicting collapse hazards in the Changbai Mountain region. The study demonstrates noteworthy innovation by integrating SHAP analysis with advanced machine learning models, providing both practical insights for risk management and valuable scientific understanding of key influencing factors. The manuscript is well-organized, methodologically sound, and highly relevant to the field. I recommend acceptance of the manuscript with minor revisions and offer the following comments.

1. For a high-quality journal like this, I recommend that the authors thoroughly review the manuscript's language and grammar, and consider having it edited by a native English speaker to ensure clarity and accuracy.

2Page 4, Table 1: Specify the temporal resolution of NDVI and average annual rainfall data. For example, clarify whether these data are derived from monthly, seasonal, or annual records. Also, include the spatial resolution (e.g., 30 m or a larger grid).

3Did you derive or develop Equation (21)? If not, please add related references to Equation (21) so that readers can check these references for further details.

4Table 4: The table numbering appears inconsistent, with both the model performance evaluation table and the hierarchical raster occupancy and frequency ratios table labeled as "Table 4." Please ensure all tables are numbered sequentially and update the text references accordingly for clarity.

54.7 Variable importance: The SHAP analysis discusses "positive and negative SHAP values" in relation to collapse risk. Add a short explanation for non-expert readers about what these values signify in practical terms.

6Please label the figure names correctly. The caption for Figure 8 mentions two sub-figures, (a) and (b), but the figure presented only corresponds to sub-figure (b) "Hazard Zoning." Ensure that all figure names and captions accurately correspond to the content presented in the figures.

7. The color contrast in Figure 9 is insufficient, and the annotated scales are unclear. Please enhance the color differentiation and clarify the scales to avoid potential misunderstandings.

8Please review all references to ensure formatting consistency according to the journal’s guidelines. Pay special attention to references with URLs and DOI links, and verify that all in-text citations correspond to entries in the reference list, with no missing or unmatched references.

Recommendation: Minor revisions required.

Reviewer 2 Report

Comments and Suggestions for Authors

In my opinion, the subject of this work is relevant to the journal readers and sufficiently novel and interesting to warrant publication. The research questions are essential for the territory's prevention planning and management. The work sound and the overall approach envisioned and implemented are generally correct. All the key elements (i.e. abstract, introduction, methodology, results, and discussion of the results) are present and arranged. The science underlying the debate is solid. Figures and tables, generally, are all necessary.

However, the introduction section requires some improvement to clearly outline the research's significance and context. The literature review appropriately gathers and discusses published works related to the subject but falls short in critically confronting and interrelating conflicting studies. This section should establish a solid theoretical foundation and identify challenges and gaps in knowledge that this research seeks to address. Furthermore, articulating the research questions and objectives is underdeveloped and needs to be more explicit. The text often assumes that the reader is more familiar with the evidence presented than is realistic to expect, and the references to the terminology used are sometimes vague (or inexistent). The authors use several terms to describe their product, such as collapse hazard, susceptibility, model validation (could be more correctly verification or performance evaluation – indeed, what the authors do is a verification of the performance of the models built and not a validation that is a very different thing in geostatistics). Fundamentally, there are ambiguities in using the term “Collapse Hazard” in the title and manuscript. Indeed, in this specific case, the correct term must be “Landslide Susceptibility”. Each of these terms must be defined in detail in the introduction so the readers aren’t confused with their use in their field because they mean something different across nations, scientific fields, and research/management. Similarly, the study area requires additional details to contextualise the rationale and importance of the study and some bibliographic references to support missing geological and geomorphological descriptions.

The authors should carefully describe data collection methods and provide a study design that aligns with the research objectives. This includes presenting all procedures logically, clearly identifying and justifying the selection of variables (landslide assessment factors), and choosing strategies and techniques. Additionally, there is confusion between the presentation of methods and some results. Multicollinearity results must be more explicit…which variables were not used in the modelling procedures after applying the multicollinearity??? And why? Why did the author divide the data into two datasets (train and test)…? Three datasets are needed for machine learning approaches: training, testing and validation.

The included figures require some improvement in graphics and cartographic quality. The legend of Figure 2 needs more information.

The discussion is overly focused on internal data analysis, with little to no comparison with existing studies or methodologies. A more substantial discussion should address the methodology's advantages, limitations, and applicability. The authors should explore whether their findings are replicable in other settings and articulate how their results differ from or align with similar studies. Additionally, they need to discuss the implications of the findings and their contribution to the field.

Finally, I strongly recommend that the authors seek the assistance of a professional English science editor to improve the manuscript's style, grammar, and sentence structure. This will ensure greater clarity and precision, making the text more accessible to readers.

While this work has some merit, it must be revised to address the identified weaknesses before it can be considered suitable for publication.

Comments on the Quality of English Language

I strongly recommend that the authors seek the assistance of a professional English science editor to improve the manuscript's style, grammar, and sentence structure. This will ensure greater clarity and precision, making the text more accessible to readers.

Reviewer 3 Report

Comments and Suggestions for Authors

An article on the possibility of predicting collapses based on the use of artificial intelligence models is submitted for review.

The study is devoted to predicting the risks of geological collapses in the Changbai Mountains region using machine learning methods. The relevance of the work is due to the need to prevent disasters that threaten human lives, infrastructure and ecosystems. The use of machine learning to analyze the complex interactions of geological, meteorological and anthropogenic factors allows us to move from reactive to proactive risk management strategies. Of particular importance is the integration of model interpretability, which increases confidence in forecasts and facilitates decision making.

The novelty of the study lies in the creation of accurate collapse risk maps for the Changbai Mountains region using optimized machine learning algorithms and interpretation of the results through SHAP analysis. The authors combine hyperparameter optimization (Particle Swarm Optimization method) with model interpretation through SHAP values. This approach is rare in geological risk studies.

Research methodology: a set of 651 collapse events and 13 risk factors, including topographic, climatic and anthropogenic parameters, were used. Multicollinearity assessment (VIF, correlation analysis) and data split into training/test sets (70/30%). Model training with hyperparameter optimization (PSO) and evaluation through metrics (Accuracy, Precision, F1, AUC-ROC). SHAP analysis to determine the contribution of factors and visualize their interactions. The methodology is systematic and meets modern standards. However, the lack of cross-validation and the limited amount of data (651 events) may reduce the reliability of the results.

The paper is important for geoscientists who apply machine learning for explaining and predicting Collapse Hazards all over the world, especially in mountains regions.

In general, the structure of paper is fine and clear. Nevertheless, there are questions to the paper:

  1. Using 30-meter resolution elevation models significantly increases the risk of missing local anomalies. It is recommended to use more accurate elevation models or justify the sufficiency of detail of the current elevation model.
  2. The sample of 651 events is small for training complex models, which may limit their generalizability. Appropriate comments must be included.
  3. It is not sufficiently clear how exactly risk maps can be integrated into the disaster management system. Appropriate comments must be included.
  4. There is no comparison with neural networks or hybrid approaches, which could strengthen the conclusions.
  5. Climate change and anthropogenic activity require taking into account time series, which is not reflected in the work.

After making the above improvements, the article can be recommended for publication.
